# Improved Vegetation Ecological Quality of the Three-North Shelterbelt Project Region of China during 2000–2020 as Evidenced from Multiple Remotely Sensed Indicators

**Chao Li** [1,2]**, Shiqiang Zhang** [1,2,]*[ID]**, Manyi Cui** [1,2]**, Junhong Wan** [1,2]**, Tianxing Rao** [1,2]**, Wen Li** [1,2] **and Xin Wang** [3]

1   College of Urban and Environmental Science, Northwest University, Xi'an 710127, China
2   Shaanxi Key Laboratory of Earth Surface System and Environmental Carrying Capacity, Northwest University, Xi'an 710127, China
3   Department of Geography, Hunan University of Science and Technology, Xiangtan 411100, China
*   Correspondence: zhangsq@nwu.edu.cn

**Abstract:** Evaluation of the long-term effect of ecosystem recovery projects is critical for future ecological management and sustainable development. The Three-North Shelterbelt (TNS) is a large-scale afforestation project in a crucial region of China. Numerous researchers have evaluated the vegetation ecological quality (VEQ) of the TNS using a single vegetation indicator. However, vegetation ecosystems are complex and need to be evaluated through various indicators. We constructed the vegetation ecological quality index (VEQI) by downscaling net primary productivity, leaf area index, fractional vegetation cover, land surface temperature, vegetation moisture, and water use efficiency of vegetation. The spatiotemporal characteristics and main contributing factors of VEQ in the TNS from 2000 to 2020 were investigated using SEN+Mann−Kendall, Hurst exponent, geographical detector, and residual trend analysis testing. The results suggest that VEQ in the TNS showed an improving trend over the 21-year study period. The areas with significant improvements were concentrated in the central and eastern parts of the TNS. Significant deterioration occurred only sporadically in various urban areas. Characteristics of future unsustainable VEQ trends could be detected across the TNS. Precipitation, vegetation type, soil type, elevation, and solar radiation exhibited the greatest impact on VEQ throughout the TNS. Human activities (e.g., afforestation and government investments) were the dominant factors and had a relative contribution of 65.24% to vegetation area change. Our results provide clues for assessing environmental recovery and sustainable development in other regions.

**Keywords:** vegetation ecological quality; Geodetector; climate variation; ecological restoration; Three-North Shelterbelt

## 1. Introduction

Vegetation has numerous functions, including air purification, water containment, climate regulation, and landscaping, playing an important role in processes such as energy transfer between the land and the atmosphere and the maintenance and optimization of ecosystem services [1,2]. Vegetation is also an essential component of terrestrial ecosystems and is a link between the atmosphere, water, organisms, rocks, soil, and other elements of the environment [2,3]. Therefore, evaluating the long-term effect of ecosystem recovery projects is critical for future environmental management and sustainable development. One region that underwent a large-scale ecosystem recovery is in the north of China (Northwest, North China, and Northeast), which was affected by serious desertification before the 1970s [4]. This region contains scarce vegetation, wood, fuel, fertilizer, and fodder. The poor natural environment severely restricts regional socioeconomic development [5]. The Chinese government planned and started implementing the Three-North Shelterbelt (TNS) project in 1978 to improve the harsh environment. The project is one of the most remarkable environmental rehabilitation projects in human history and involves returning pastures to

grassland, planting trees, and reverting farmland to forest and closed hillsides to facilitate afforestation.

Vegetation ecological quality (VEQ) can reflect the vegetation function of a terrestrial ecosystem, which is the core guarantee to achieve the goal of "double carbon" [6,7]. After decades of construction, the VEQ of the TNS region has changed considerably [8], and its sustainability has received great attention [9,10]. Climatic factors—temperature (TEM), precipitation (PRE), potential evapotranspiration (EVA), and wind speed (WS)—are also important drivers of vegetation change in the TNS, particularly under global warming [8,11]. Therefore, it is necessary to evaluate the long-term impact of the TNS project on vegetation in regards to climate variation and anthropogenic impacts.

The impacts of climate variation and anthropogenic activities on the vegetation dynamics of the TNS have been evaluated primarily based on single remotely sensed indicators, such as the normalized differential vegetation index (NDVI) [8,12], the fractional vegetation cover (FVC) [13], the leaf area index (LAI) [11], net primary productivity (NPP) [14], water use efficiency (WUE) [15], or gross primary productivity (GPP) [16]. However, a single indicator can only one-sidedly reflect changes in VEQ in a particular dimension and may lead to potential bias. Vegetation ecosystems are complex and understanding changes in their characteristics requires a combination of multiple indicators [3,17].

Composite indicators of the VEQ can be composed entirely of remote sensing data. The remote sensing-based ecological index (RSEI) proposed by Xu [18] integrates multiple factors, including wetness (WET), dryness, greenness, and heat. RSEI can be used indirectly to carry out a comprehensive evaluation of regional VEQ. This composite indicator, while widely used [19–21], is based on Landsat data. Due to its spatial resolution, it is limited to monitoring and evaluating the VEQ of small-scale areas and cannot be easily applied to large-scale areas. Another category is the comprehensive evaluation model, constructed by integrating remote sensing data and other data, such as the ecological index (EI) [22], introduced by the State Ministry of Environmental Protection of China, which uses remote sensing, ground monitoring, and statistical data. However, geographic, scale, and time constraints often prevent complete data availability. Additionally, the fusion with statistical data makes reaching the pixel scale difficult. Therefore, the model still needs to be improved to evaluate long-term regional vegetation change. Li et al. [6] constructed the vegetation ecological quality index (VEQI) based on six remote sensing data and monitored the VEQ on a large scale, which provides a reference basis for the construction of an assessment indicator for the TNS in this study.

According to the above research gaps, the objectives of this study were: (1) to construct the VEQI by employing multiple indicators of vegetation using pixel-scale remote sensing data; (2) to analyze the spatiotemporal evolution characteristics of VEQ in the TNS since the beginning of the 21st century and to forecast the future trends of VEQ changes; and (3) to quantify the relative contributions of climate variation and anthropogenic impacts on the VEQ in the TNS. This study has important ecological and socio-economic significance for monitoring the dynamic VEQ changes in the TNS.

## 2. Materials and Methods

### 2.1. Research Area

The TNS spans 13 provinces (autonomous regions and municipalities), with a total area of about 4.07 million $km^2$ (Figure 1) [23]. The terrain consists mainly of mountains, hills, plains, and plateaus (Figure 1). Due to the vast territory, PRE decreases from south to north and east to west, with most areas receiving less than 600 mm/yr (Figure 2). The average annual TEM varies in most regions between 2 °C and 5 °C (Figure 2). The local climate varies from humid in the east to semi-arid and arid in the west and can be mainly classified as an arid zone (ADZ), a semi-arid zone (SAZ), a dry semi-humid zone (DSH), a wet semi-humid zone (WSH), and a humid zone (HDZ), based on aridity data from the Resource and Environmental Science Data Center (https://www.resdc.cn/ (accessed on 3 January 2022)). The main vegetation types influenced by climate include grassland

(41.19%), agricultural land (15.75%), and forests (8.47%). Barren or sparse vegetation cover one-third of the area (35.52%) [9]. China's four sandy lands, eight largest deserts, and the wide Gobi Desert are distributed in the TNS, of which the sandy areas account for 85% of the country's total sandy land area [24]. Therefore, the implementation of the TNS project plays a crucial environmental restoration role.

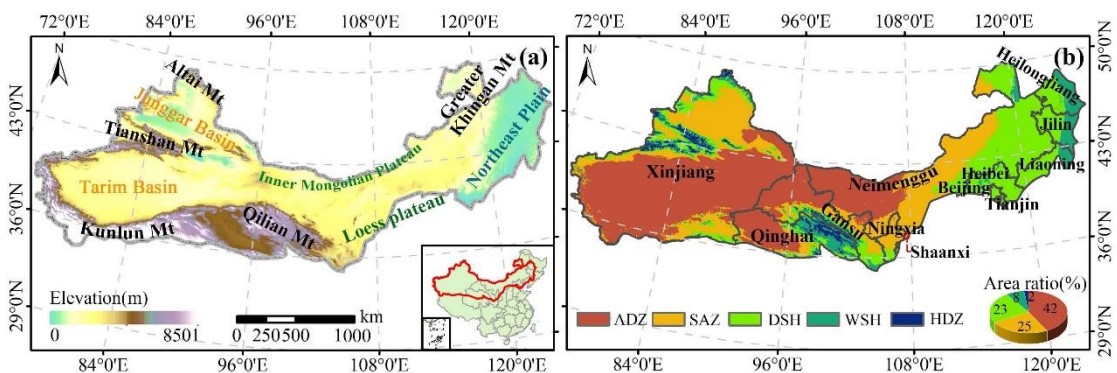

**Figure 1.** Topographic (**a**) and climatic zoning (**b**) of the study area.

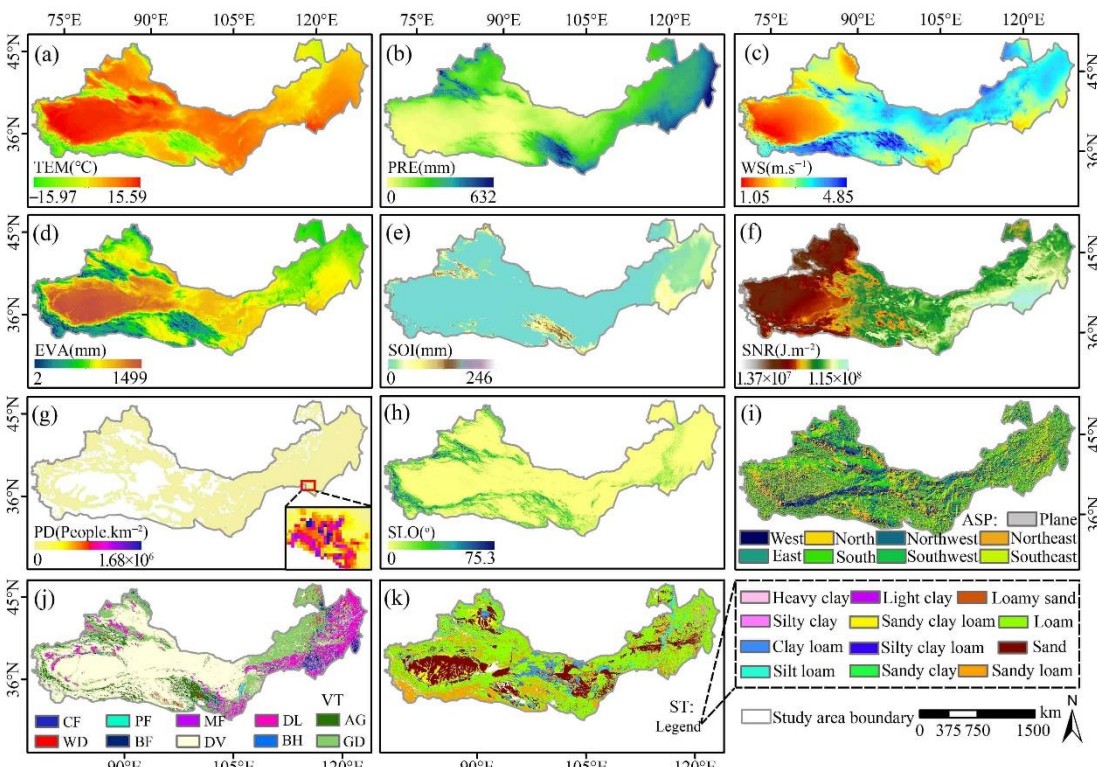

**Figure 2.** Spatial distribution of the 11 drivers of VEQI. CF, PF, MF, DL, AG, WD, BF, DV, BH, and GD in (**j**) represent coniferous forest, paddy field, mixed farms, dry land, alpine grassland, wetland, broad-leaved forest, desert vegetation, and bush and grassland, respectively.

### 2.2. Material

MODIS and GLASS data were obtained from NASA (https://search.earthdata.nasa.gov (accessed on 5 November 2021)) and the National Center for Earth System Science Data of China (http://www.geodata.cn/ (accessed on 22 November 2021)), respectively (Table 1). After format conversion, reprojection, and resampling of the above MODIS data products using the MODIS Reprojection Tool (MRT) software [25], all raster data were subjected to image mosaicking and cropping using ENVI 5.3 software. The MODIS and GLASS series data have anomalous or missing values due to the influence of sensor observation angle,

solar illumination angle, atmospheric aerosols, rain, and snow. Therefore, the outliers need to be processed. The main steps were to set the anomalies as NoData in ArcGIS software, and then fill the NoData by linear interpolation. The Savitzky–Golay filter [26] was also used to remove the residual noise to generate MODIS and GLASS data from the time series.

**Table 1.** Remote sensing products used for the study.

| Remote Sensing Products | Representative Data | Spatiotemporal Resolution | Used for Calculation of Evaluation Indexes |
|---|---|---|---|
| GLASS ET | Surface Evapotranspiration | 1000 m–8 Day | WUE |
| MOD09A1 | Surface Reflectance | 500 m–8 Day | WET |
| MOD17A2H | Gross Primary Production | 500 m–8 Day | WUE |
| MOD15A2H | Leaf Area Index | 500 m–8 Day | LAI |
| MOD13A3 | Normalized Differential Vegetation Index | 1000 m–Monthly | FVC |
| MOD11A2 | Land Surface Temperature | 1000 m–8 Day | LST |
| MOD17A3 | Net Primary Production | 500 m–8 Day | NPP |
| MCD12Q1 | Land Cover Type | 500 m–Yearly | – |

MOD11A2 and MOD15A2H were processed to obtain annual LST and LAI data, respectively. We used the pixel dichotomy method [6] to process MOD13A3 to obtain the monthly FVC data and performed raster operations to obtain the annual FVC data. We performed raster calculations for GLASS ET, MOD17A2H, and MOD17A3 to obtain annual surface evapotranspiration, as well as GPP and NPP data, respectively. WUE indicates the ratio of water loss per unit of carbon uptake in vegetation ecosystems, which can measure the vegetation growth status [27]. It can be calculated from the ratio of GPP to surface evapotranspiration [27]. The WET of vegetation is an important component of the plant body and is a reactive substance in the metabolic process of vegetation [28]. We first used MOD09A1 and the modified MODIS tassel cap conversion formula [6] to calculate the vegetation WET at 8-day temporal resolution, and then performed the mean operation to obtain the annual WET data. Because water, bare soil, and snow cover have no value, they were ignored by the masking method. A mask file can be obtained through land cover type data (MCD12Q1).

The monthly-scale PRE [29], TEM [30], and EVA [31] data (Figure 2), with a spatial resolution of $0.0083° \times 0.0083°$ from 2000 to 2020, were obtained from the National Tibetan Plateau Data Center (http://data.tpdc.ac.cn/ (accessed on 7 February 2022)). Solar net radiation (SNR) data were downloaded from the ERA5-Land monthly dynamic reanalysis dataset at $0.1° \times 0.1°$ resolution (https://cds.climate.copernicus.eu/ (accessed on 9 February 2022)). The monthly WS and soil water (SOI) data, with a spatial resolution of about 4 km (1/24 degree), were obtained from the TerraClimate dataset (https://www.climatologylab.org/ (accessed on 9 February 2022)), which has been widely used in hydrology and ecology studies [32,33].

The digital elevation model (DEM) [34], soil type (ST) [35], and vegetation type (VT) [36] data for China, with a spatial resolution of 1 km (Figure 2), were downloaded from the National Cryosphere Desert Data Center (http://data.casnw.net/portal/metadata/ (accessed on 11 February 2022)). The spatial distribution of slope (SLO) and slope aspect (ASP) were derived using ArcGIS software. The VT includes seven kinds of natural vegetation (coniferous forests, broad-leaved forests, shrublands, alpine grasslands, grasslands, desert vegetation, and wetlands) and three types of cultivated vegetation (paddy fields, drylands, and mixed farmlands). Population density (PD) data were downloaded from WorldPop (https://www.worldpop.org/ (accessed on 12 February 2022)). Data on afforestation areas and fixed investment in forestry projects were obtained from the Chinese Forestry Statistical Yearbook for various years.

### 2.3. Methodology

The VEQI was constructed by principal component analysis (PCA) [19] based on six indicators (Figure 3). The spatio-temporal patterns and future trends of the VEQ were studied using SEN+Mann-Kendall [37–39] and Hurst exponent testing [40]. The effects of single-factor and two-factor interactions on the VEQ were analyzed using geographic detectors [41]. Finally, the driving mechanism of VEQ was quantified using the residual analysis method [42]. The workflow is shown in Figure 3.

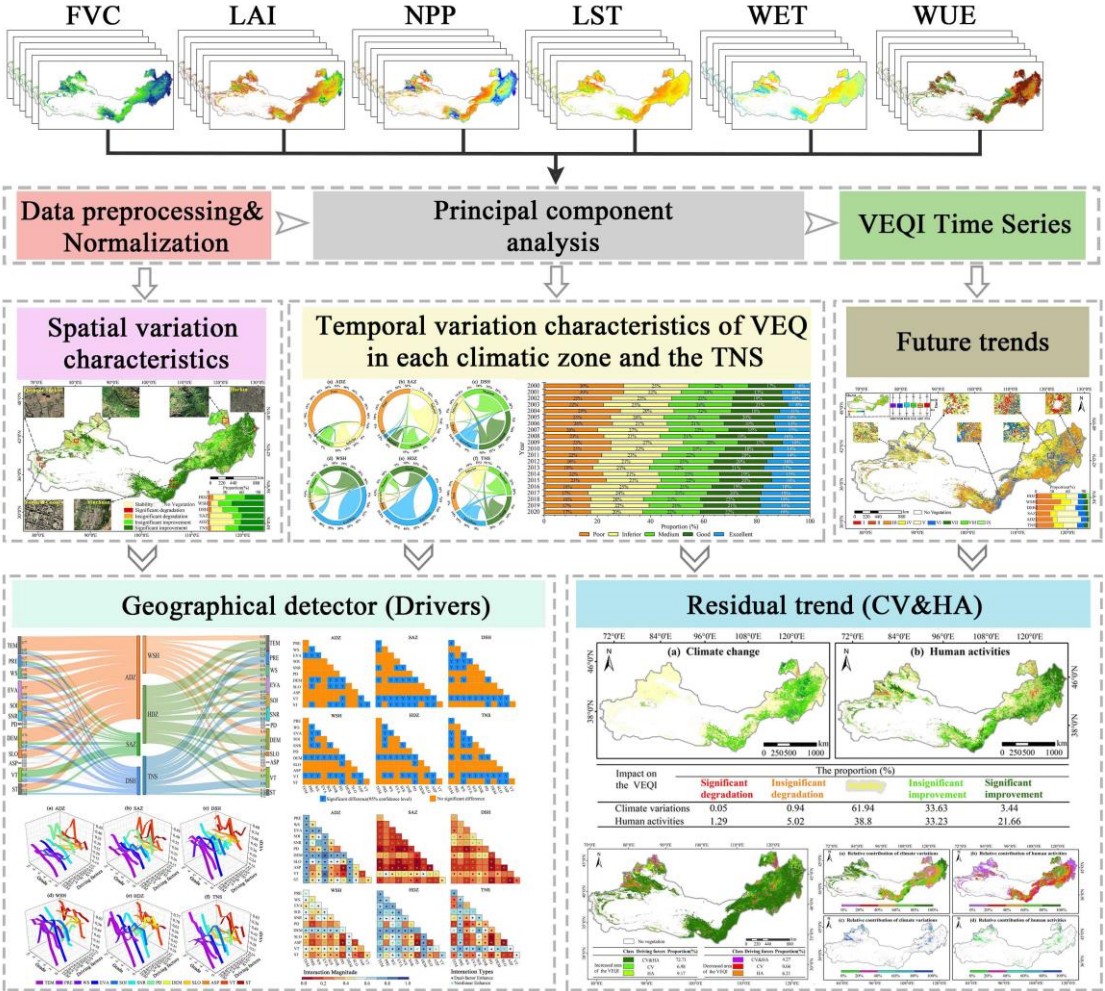

**Figure 3.** Research flowchart.

### 2.3.1. Construction of VEQI and Trend Analysis

We used FVC, LAI, NPP, WET, LST, and WUE as the basic evaluation indexes of the proposed VEQI [6]. PCA was used to downscale the six indicators. The complete description of VEQI calculation using PCA can be found in the article by Li et al. [6]. The results of PCA for four years are presented in this paper (Text S1 and Table S1). The final VEQI was normalized to range between [0,1] [43]. The closer the VEQI value is to 1, the better the VEQ. We also used the natural interval method to classify the VEQI values into five classes (poor, inferior, medium, good, and excellent). We used SEN-slope to analyze the trend of the VEQI in the TNS over the past 21 years. In addition, the Mann–Kendall method was used to perform a significance test of the VEQI trend [44,45]. The sustainable characteristics of VEQI were analyzed using the Hurst exponent [46]. The complete description of the SEN slope, Mann–Kendall test, and Hurst exponent can be found in the article by Liu et al. [47]. In this study, the SEN, Mann–Kendall, and Hurst exponent evaluations were also superimposed to project future trends of VEQI.

### 2.3.2. Geodetector

We analyzed the spatial heterogeneity of VEQ and the driving forces using Geodetector, which is highly versatile and includes the following four detectors:

Factor detector: used to measure the explanatory power of the relevant factors for changes in VEQ. The explanatory power of each factor is measured by the $q$ [41,48,49]:

$$q = 1 - \frac{\sum_{h=1}^{L} N_h \sigma_h^2}{N \sigma^2} \tag{1}$$

where $h$ is the stratification of variable Y or factor X; $N_h$ and $N$ are the number of cells in stratum $h$ and the whole region, respectively; $\sigma_h^2$ and $\sigma^2$ are the variance of Y values in stratum $h$ and the whole region, respectively. The value range of $q$ is [0,1]. The closer $q$ is to 1, the stronger the explanatory power of the X on Y.

Ecological detector and risk detector: The ecological detector determines whether there is a significant difference in the effect of any two factors on the spatial distribution of VEQI [41]. The risk detector can determine whether there is a significant difference in the mean values of attributes between two sub-regions [41].

Interaction detector: The interaction detector is used to identify the interaction between the two drivers. The interaction results show whether the two variables are weakening or enhancing each other or are causing VEQI changes independently of each other. The complete description of the Geodetector can be found in the article by Wang et al. [41].

It is clear from the conception and principle of Geodetector that it can only process discrete variables [41,48]. Since the driver variables we used contained continuous variables, the discrete processing of continuous variables was required. We performed raster mean calculations on the time series data of TEM, PRE, WS, EVA, SOI, SNR, and PD to obtain multi-year mean data. The ASP, VT, and ST data are type variables and do not need to be discretized. Finally, the above data were entered into the Geodetector model to obtain the results.

### 2.3.3. Residual Analysis

We used residual analysis [42] to quantify the influences of climate change and human activities on VEQ. The steps include: (1) establishing a binary linear regression model for each pixel with TEM, PRE, and SNR data of the time series from 2000 to 2020 as independent variables and VEQI of the corresponding year as a dependent variable and calculating the parameters in the model; (2) obtaining the predicted $VEQI_{CV}$ based on the calculated regression model parameters and climate data to quantify the impact of climate factors on VEQI; (3) performing the difference analysis between the observed $VEQI_{obs}$ and the predicted $VEQI_{CV}$ to obtain the residual $VEQI_{HA}$ to quantify the contribution of human activities to VEQI. The calculation equation is shown as follows:

$$VEQI_{CV} = a \times Tem + b \times Pre + c \times Solr + d \tag{2}$$

$$VEQI_{HA} = VEQI_{obs} - VEQI_{CV} \tag{3}$$

where $VEQI_{CV}$ and $VEQI_{obs}$ refer to the predicted VEQI and calculated true VEQI, respectively; a~d are the regression parameters; $CV$ and $HA$ denote climate variation and human activities, respectively; *Tem*, *Pre*, and *Solr* refer to mean annual temperature, accumulated precipitation, and solar radiation, respectively. We further divided the main drivers of VEQI change, according to Table 2 [48], and calculated the relative contribution of climate variation and human activities to VEQI change [48,50–52].

**Table 2.** Determination criteria of drivers of VEQI changes and the relative contribution of climate variation (CV) and human activities (HA).

| $Slop_{obs}$ | Driving Factor | Standard of Division | | Relative Roles (%) | |
|---|---|---|---|---|---|
| | | $Slop_{CV}$ | $Slop_{HA}$ | Climate Variation | Human Activities |
| >0 | CV&HA | >0 | >0 | $Slop_{CV}/slop_{obs}$ | $Slop_{HA}/Slop_{obs}$ |
| | CV | >0 | <0 | 100 | 0 |
| | HA | <0 | >0 | 0 | 100 |
| <0 | CV&HA | <0 | <0 | $Slop_{CV}/Slop_{obs}$ | $Slop_{HA}/Slop_{obs}$ |
| | CV | <0 | >0 | 100 | 0 |
| | HA | >0 | <0 | 0 | 100 |

Notes: $Slop_{obs}$, $Slop_{CV}$, and $Slop_{HA}$ refer to slopes of actual observed VEQI, predicted VEQI, and residual VEQI, respectively.

## 3. Results

### 3.1. Trends of VEQ in Each Climatic Zone of TNS

The temporal trends of VEQ by different climate zones and in the TNS suggest an improving trend (Figure 4a–c), except in ADZ. Moreover, the trends passed the 99% significance test ($p < 0.05$), indicating that over the past 21 years, the average VEQ in the TNS has been greatly improved. The most significant improvement was observed in the WSH and DSH climatic zones, with an average annual growth rate of 0.0036 and 0.0027, respectively. On the other hand, ADZ improved only with a yearly growth rate of 0.0001. Spatially, 68.93% of the areas showed an improving trend in VEQ (Figure 4g), while 26.34% of the areas were stable, and only 4.73% showed a deteriorating trend. Significantly improved areas (45.26%) were concentrated in the central and eastern parts of the TNS, namely the Qilian Mountains, Loess Plateau, Junggar Basin, Inner Mongolia Plateau, Northeast Plain, and Tarim Basin. Significantly deteriorated areas (1.07%) did not show a large regional clustering, but were mainly scattered in various urban areas that were primarily influenced by the accelerated urbanization process, such as in Harbin, Qeshqer Sehri, and Yarkant County (Figure 4g).

It can be noted that the proportion of inferior, medium, and good VEQ did not fluctuate greatly (Figure S1). Instead, the proportion of poor VEQ decreased significantly (13%), and the proportion of excellent VEQ increased significantly (13%). The spatial transfer matrix of VEQ was obtained based on the Markov transfer algorithm [7,53,54]. The increase in the area of good and excellent VEQ in the TNS was mainly derived from the transformation of the next level VEQ, while the decrease in the VEQ of the other three areas was due to their transformation to more advanced types (Figure 5). The transfer characteristics of different VEQ types in each climate zone were similar to those of the TNS, but there were some differences in the transfer characteristics of VEQ types among different climate zones. For example, 30.73% of the WSH had a VEQ type changing from good to excellent, while the good type did not transfer to excellent in ADZ.

The results show that the Hurst exponent of VEQ in the TNS and each climate zone were less than 0.5 (Figure 6), and the area proportions of the TNS showing persistent and non-persistent characteristics were 28% and 72%, respectively. This indicates that the VEQ in most areas of the TNS will be less sustainable in the future. The results of the SEN slope, Mann–Kendall test, and Hurst exponent were superimposed to project the future trend of VEQ (Table S2, Figure 6). The results show that the VEQ of the TNS will continue to improve, remain stable and deteriorate in the future, accounting for 22.35%, 26.34%, and 51.31%, respectively, which indicates that the VEQ of the TNS may deteriorate in the future.

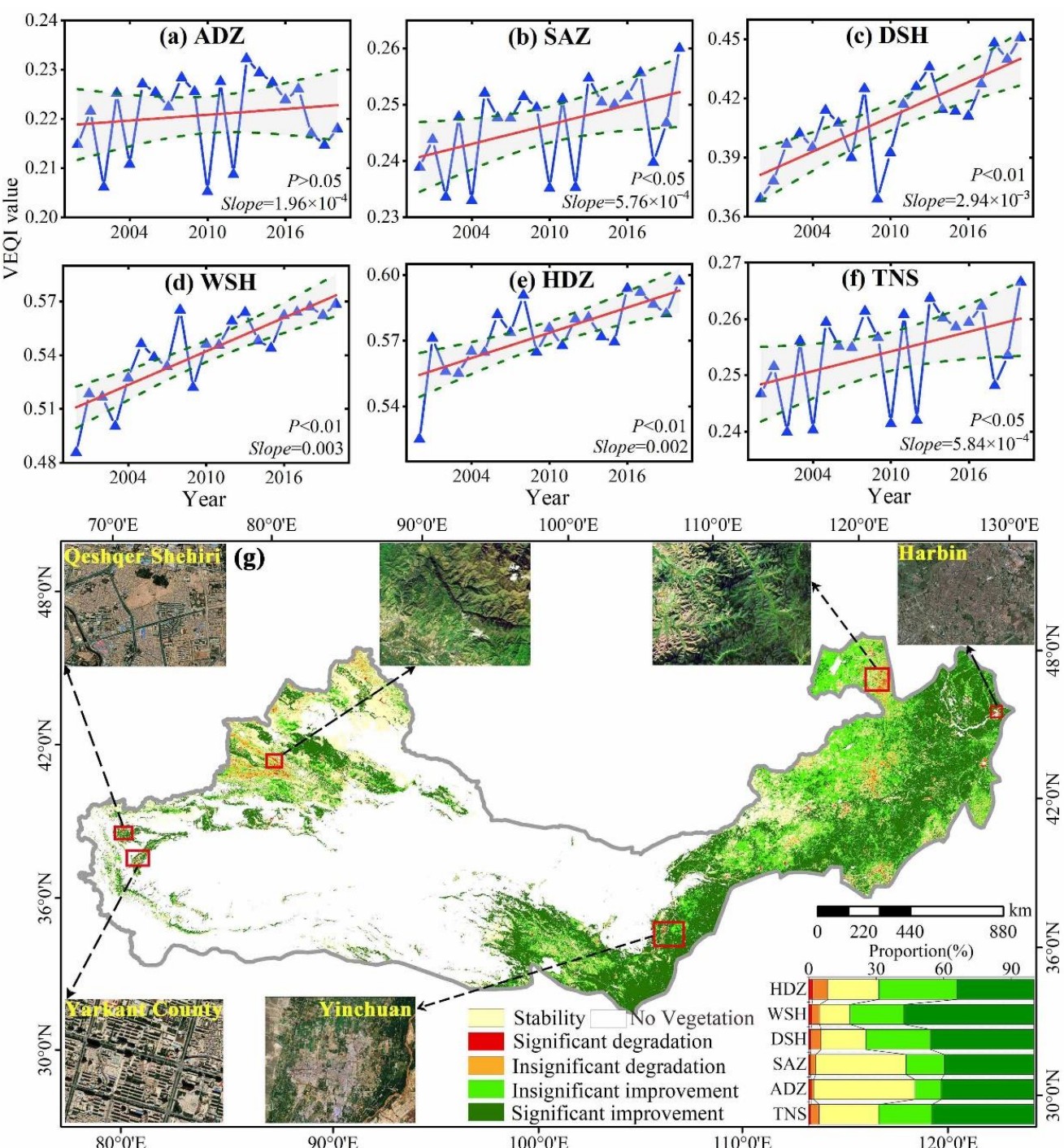

**Figure 4.** Temporal and spatial trends of VEQ in the TNS and each climatic zone: (**a–f**) the temporal trends of VEQ in TNS and each climate zone; (**g**) the spatial trend of VEQ in TNS.

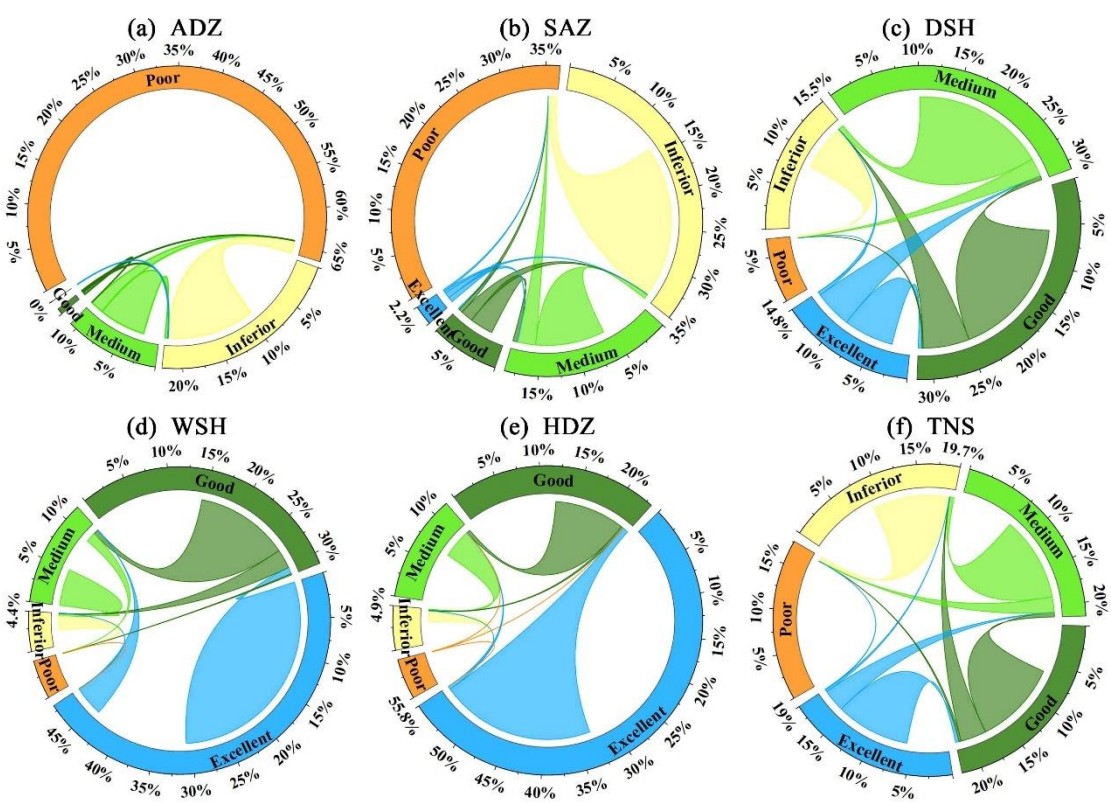

**Figure 5.** Proportion of transferred area of each type of VEQ in the TNS (**f**) and each climatic zone (**a**–**e**).

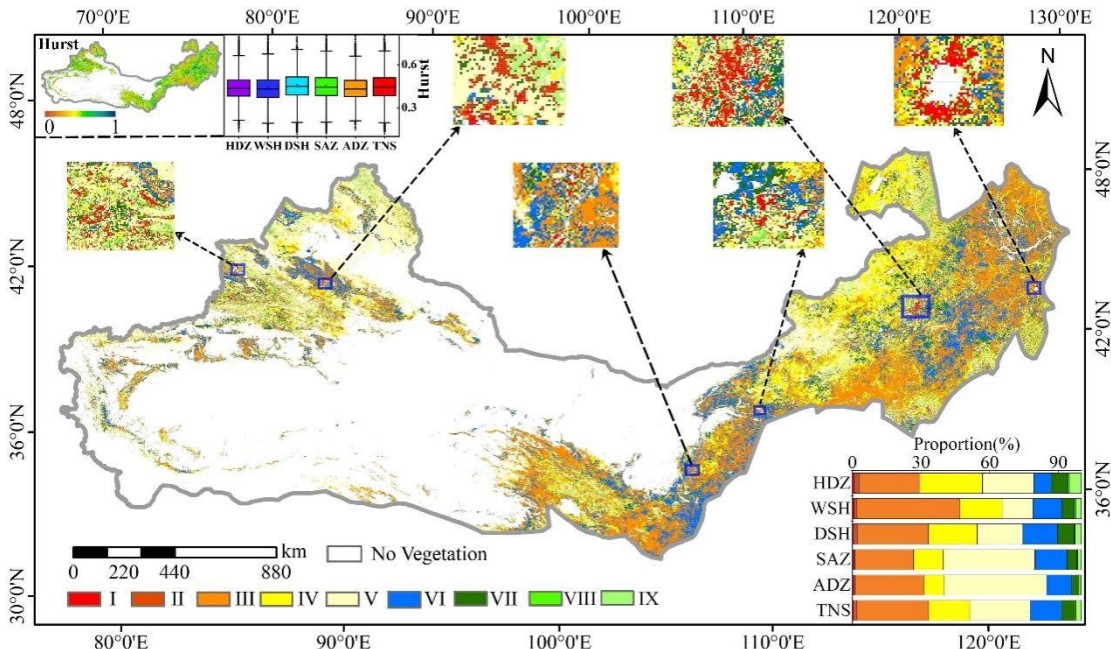

**Figure 6.** Future trends of VEQ in the TNS. (I-IX denote the different future trends shown in Table S2).

### 3.2. The Factors Influencing the Geographic Distribution of VEQ

We analyzed the impact of 12 factors (TEM, PRE, WS, EVA, SOI, SNR, PD, DEM, SLO, ASP, VT, and ST) on the change of VEQI using the factor detector in the Geodetector model. The impact of each factor on the change of VEQI in the TNS was significantly different ($p <$ 0.01). The contributions, in descending order, were PRE (0.60), VT (0.56), SOI (0.47), DEM

(0.32), SNR (0.31), EVA (0.19), TEM (0.14), ST (0.12), WS (0.10), PD (0.02), SLO (0.01), and ASP (0.01) (Figure 7).

**Figure 7.** Contribution of drivers to VEQI in the TNS and each climate zone.

The impact of each driver on VEQI in each climate zone was further analyzed (Figure 7), showing that the top three factors with the greatest influence were TEM, EVA, and DEM in ADZ, DEM, and EVA; VT in WSH, VT, and DEM; PRE in SAZ, DEM, and TEM; EVA in HDZ; and VT, SOI, and SNR in DSH, respectively. This indicates that individual factors exhibited different effects on each climate zone in the TNS, and that different climate subregions were also affected by various driving factors. We also found that the contribution of the 12 factors in the five climatic zones in descending order were ADZ (0.37), HDZ (0.36), WSH (0.23), DSH (0.12), and SAZ (0.10). This indicates that VEQI in the climate zones of ADZ and HDZ were more susceptible to the 12 drivers, while the 12 indicators had the least effect on SAZ.

The ecological detector was used to determine whether there was a significant difference in the effect of the two drivers on the spatial distribution of VEQI (Figure 8). For the TNS, the highest percentage of VT was significantly different from the other factors. In contrast, the lowest percentage of PRE was significantly different from the other factors. The VT was significantly different from all other factors, except that there was no significant difference with PRE. PRE was not significantly different, except for significant differences with TEM. The variability among the factors also varies among climate zones.

We analyzed the interaction contribution of 12 driving factors to the changes of VEQ in the TNS over the 21 years. There was a significant joint enhancement among the 12 factors, and no factors acted independently, which further illustrates the dominant role of each influencing factor on the VEQ (Figure 9). For the TNS, the top three interaction factors with the strongest contribution to VEQI changes were PRE ∩ DEM (0.771), PRE ∩ VT (0.745), and PRE ∩ EVA (0.744). By climate zone, the top three interacting factors that contributed to the VEQI variation in ADZ were EVA ∩ DEM, EVA ∩ VT, and TEM ∩ DEM; for DSH, they were SOI ∩ SNR, SNR ∩ DEM, and SOI ∩ VT; for HDZ, they were, in order, SOI ∩ DEM, PRE ∩ DEM, and TEM ∩ DEM; for SAZ, they were, in order, VT ∩ DEM, VT ∩ PRE, and VT ∩ TEM; and for WSH, they were, in order, SNR ∩ DEM, EVA ∩ DEM, and TEM ∩ DEM. This indicates that the same interacting factor influenced the VEQ of each climate zone differently. In addition, we found that the interactions between individual factors and other factors in regards to VEQ in the TNS were, in descending order, PRE > VT > SOI > DEM > SNR > EVA > TEM > WS > ST > SLO > PD > ASP, indicating that the interaction between PRE and other factors had the greatest effect on VEQ, while the interaction between ASP and other factors had the least effect on VEQ.

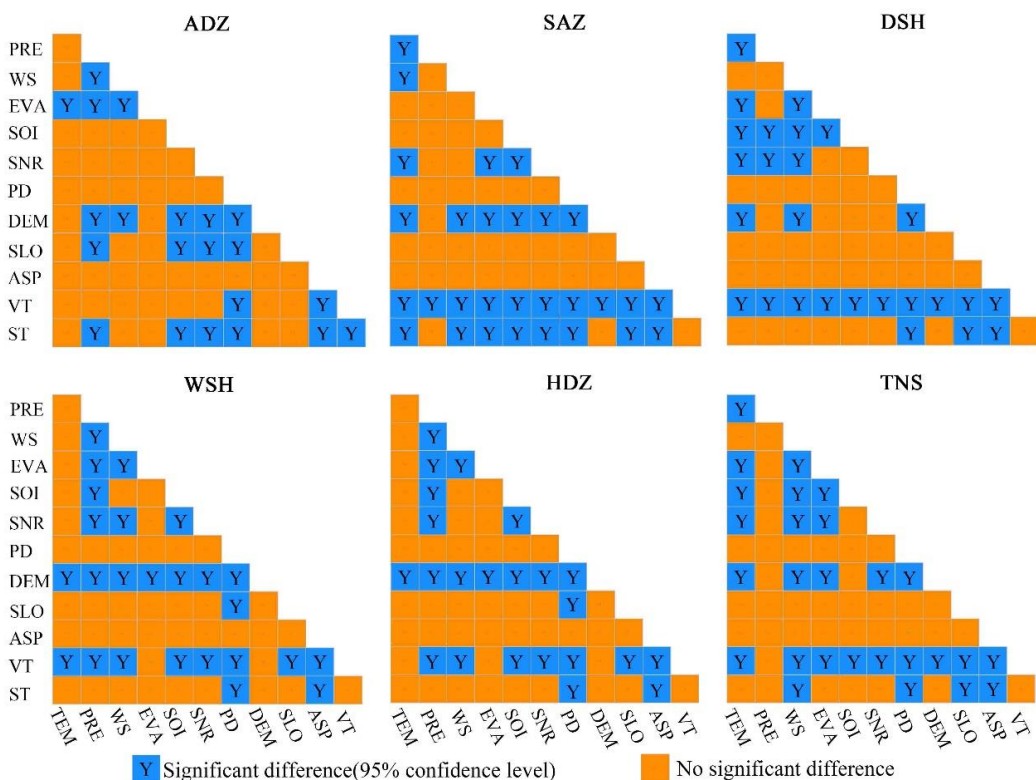

**Figure 8.** Significance test between drivers in the TNS and each climate zone.

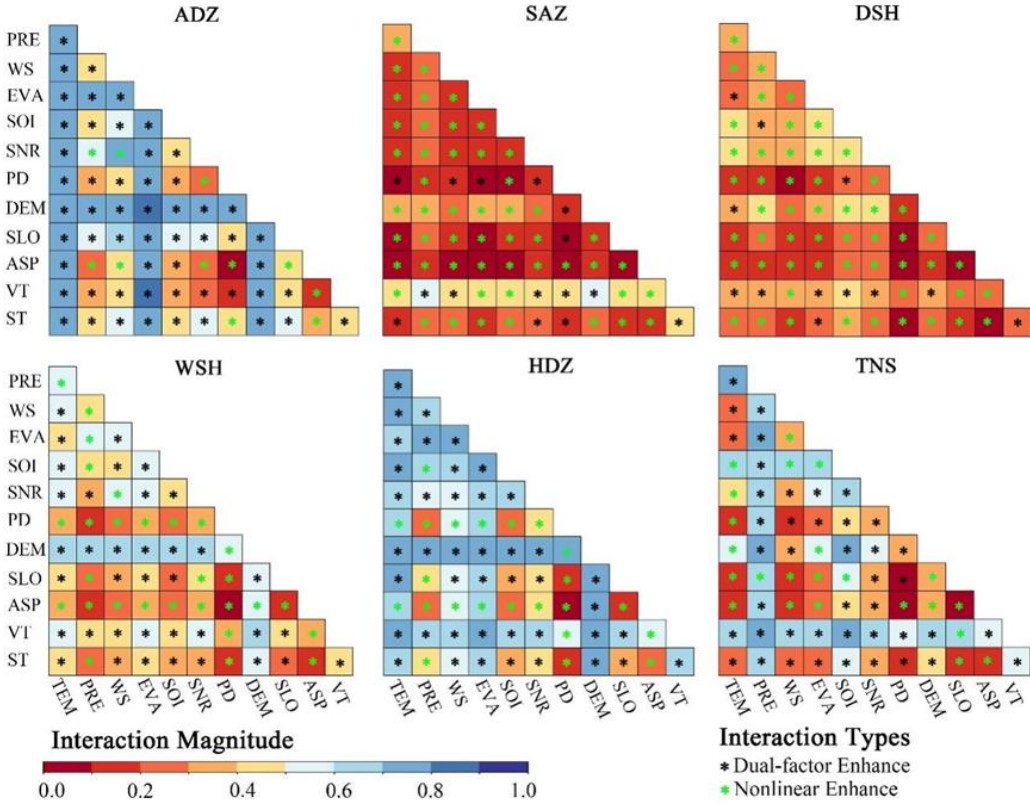

**Figure 9.** Effect of two-factor interaction on the change of VEQI in the TNS and each climate zone.

The optimal types and range of drivers favoring the maintenance of VEQ in the TNS and each climate zone were analyzed based on the risk detector (Figure 10, Table S3). For the TNS, TEM was positively related to VEQI before level 5, then negatively related to VEQI

(Figure 10). VEQI showed an increasing trend with increasing PRE. WS was positively correlated with VEQI before level 4, and then negatively correlated with VEQI. EVA was positively correlated with VEQI before level 5, then negatively correlated with VEQI. SOI was positively correlated with VEQI before level 4, then negatively correlated with VEQI, and positively correlated again starting at level 8. SNR was overall positively correlated with VEQI before level 5. The increase in PD did not cause a significant effect on VEQI.

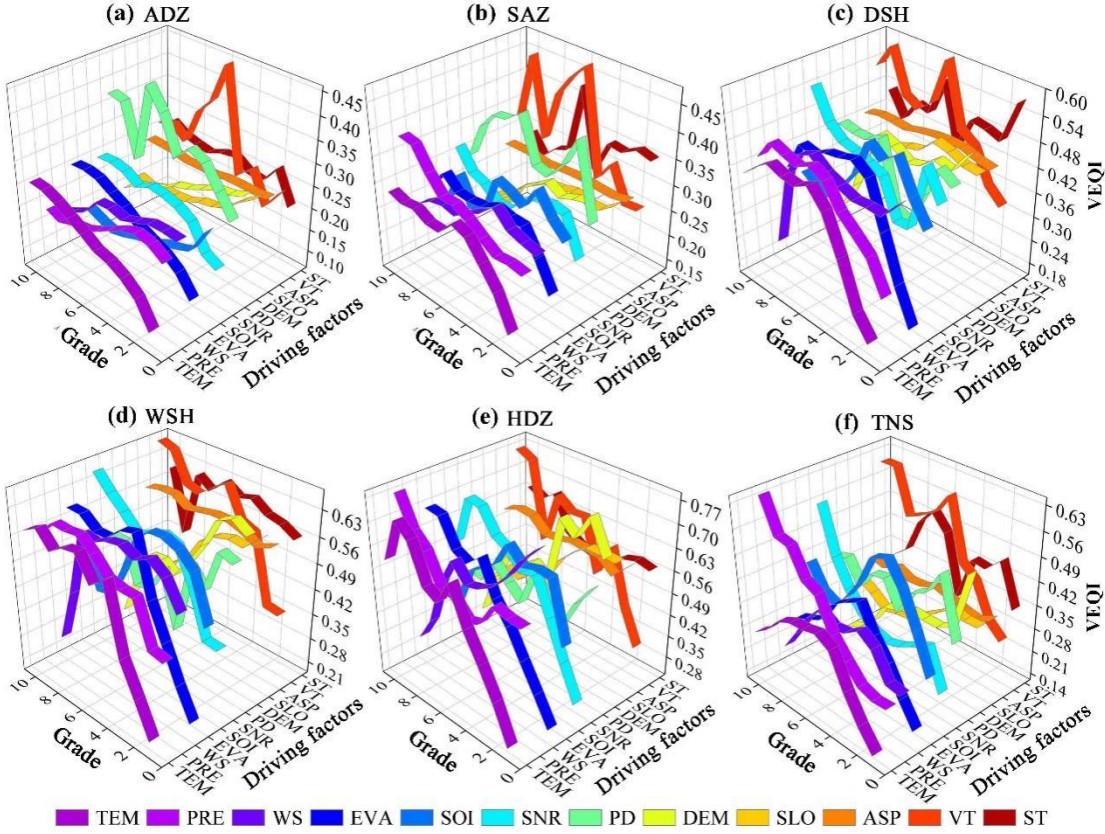

**Figure 10.** Statistical results of VEQI in the TNS (**f**) and each climate zone (**a–e**) at each level of influence.

The VEQI tended to decrease monotonically with increasing altitude. SLO was not monotonically correlated with VEQI before level 3, then negatively correlated with VEQI, and positively correlated starting at level 8. The VEQ was the best for coniferous forest and chalky sandy loam and the worst for desert vegetation and clay loam. High TEM, WS, and EVA do not promote vegetation growth, but rather inhibit it. The increase in PRE and SNR was beneficial to vegetation. VEQ deteriorated with increasing altitude and SLO. SOI, PD, and ASP did not correlate linearly with VEQ. VT and ST also affected VEQ to some extent, and the maximum VEQI values varied widely among VT and ST. In addition, we found that the VEQ trends in each climatic zone with respect to each driver were consistent with those of the TNS.

### 3.3. Impacts and Contributions of Climate Variation and Human Activities on VEQI Change

3.3.1. Impacts of Climate Variation on VEQI Change

The climate of the TNS underwent a general trend of "warming and wetting" over the 21-year study period (Figure S2). PRE and TEM showed slightly increasing trends (1.72 mm/yr and 0.004 °C/yr, respectively). SNR also showed an increasing trend of $2 \times 10^4$ J/yr. Climate variation had little effect on VEQ change in 61.94% of the TNS (Figure 11a). Climate variation improved the VEQ in 37.07% of the TNS, mainly in the Qilian Mountains, Loess Plateau, eastern Inner Mongolia Plateau, Greater Khingan Mountains,

and Northeast Plain. The 0.99% TNS where climate variation had an inhibitory effect on the VEQ of TNS was located in the central Greater Khingan Mountains. In addition, the results also suggest that climate variation has different impacts on the VEQ in different climatic zones, with the greatest differences in the ADZ and DSH climatic zones.

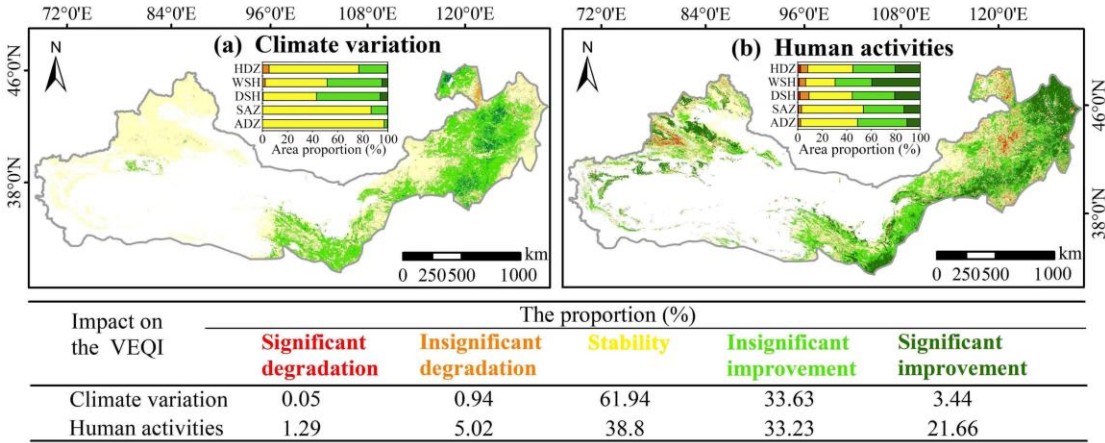

| Impact on the VEQI | The proportion (%) | | | | |
|---|---|---|---|---|---|
| | **Significant degradation** | **Insignificant degradation** | **Stability** | **Insignificant improvement** | **Significant improvement** |
| Climate variation | 0.05 | 0.94 | 61.94 | 33.63 | 3.44 |
| Human activities | 1.29 | 5.02 | 38.8 | 33.23 | 21.66 |

**Figure 11.** Impacts of climate variation and human activities on VEQI: (**a**) impact of climate variation on VEQI; (**b**) impact of human activities on VEQI.

### 3.3.2. Impacts of Human Activities on VEQI

The effects of human activities on VEQI were analyzed using the residual analysis method. The influence of human activities on the change of VEQI in the TNS and each climate zone has gradually increased (Figure S3). In particular, human activities played a significant positive role in improving VEQI in the region in the last decade. More than half (54.89%) of the areas where human activities promoted the VEQI were distributed evenly in the TNS (Figure 11b). Human activities have inhibited the VEQI in only 6.31% of the TNS (the Tianshan Mountains and the southern Greater Khingan Mountains). There were large differences in the impact of human activities on the SAZ and WSH climate zones.

### 3.3.3. Dominant Factors of VEQI Changes and Relative Contributions

We quantified the drivers of VEQI changes and their contributions as calculated by the residual analysis method in Table 2. The area of VEQI change mainly caused by climate variation accounted for 7.64% (Figure 12), and the area of VEQI increase dominated by climate variation (6.98%) was concentrated in the western part of the Junggar Basin and the Greater Khingan Mountains. The area of VEQI decrease dominated by climate variation (0.66%) was concentrated in the Greater Khingan and the Altai Mountains. The area of VEQI change dominated by human activities accounted for 15.38% (Figure 12), and the area of VEQI increase dominated by human activities (9.17%) was concentrated in the Northeast Plain. VEQI dominated by human activities (6.21%) was concentrated in the Tianshan Mountains. We also found differences in the interactive and independent effects of climate variation and human activities on the VEQI in different climate zones.

The combination of climate variation and human activities resulted in changes of VEQI in 76.98% of the TNS vegetation zone (Figure 12). We further clarified the relative contribution of climate variation and human activities to VEQI change based on the relative contributions of different drivers to VEQI changes and the original VEQI trends. About 72.71% of the vegetation regions showed the combined impact of climate variation and human activities as drivers of VEQI increase, with the average relative contributions of climate variation and human activities being 37.13% and 62.87%, respectively (Figure 13a,b). The regions with a contribution of climate variation greater than 80% accounted for 29.19%, mainly distributed in the vegetation area of Xinjiang and the northeastern Northeast Plain. The regions with the relative contribution of human activities greater than 80% accounted for 10.47%, concentrated in the Qilian Mountains region and its eastern area, the Greater

Khingan Mountains and the Northeast Plain. About 4.27% of the TNS also showed that the interaction of climate variation and human activities caused the VEQI reduction, with the average relative contribution of climate variation and human activities of 15.59% and 84.41%, respectively (Figure 13c,d). The regions with a contribution of climate variation greater than 80% accounted for 8.44% and were mainly situated in the Tianshan Mountains and the Northeast Plain. The area of the region where the relative contribution of human activities was greater than 80% accounted for 0.84% and was mainly distributed in the Greater Khingan and Altay Mountains.

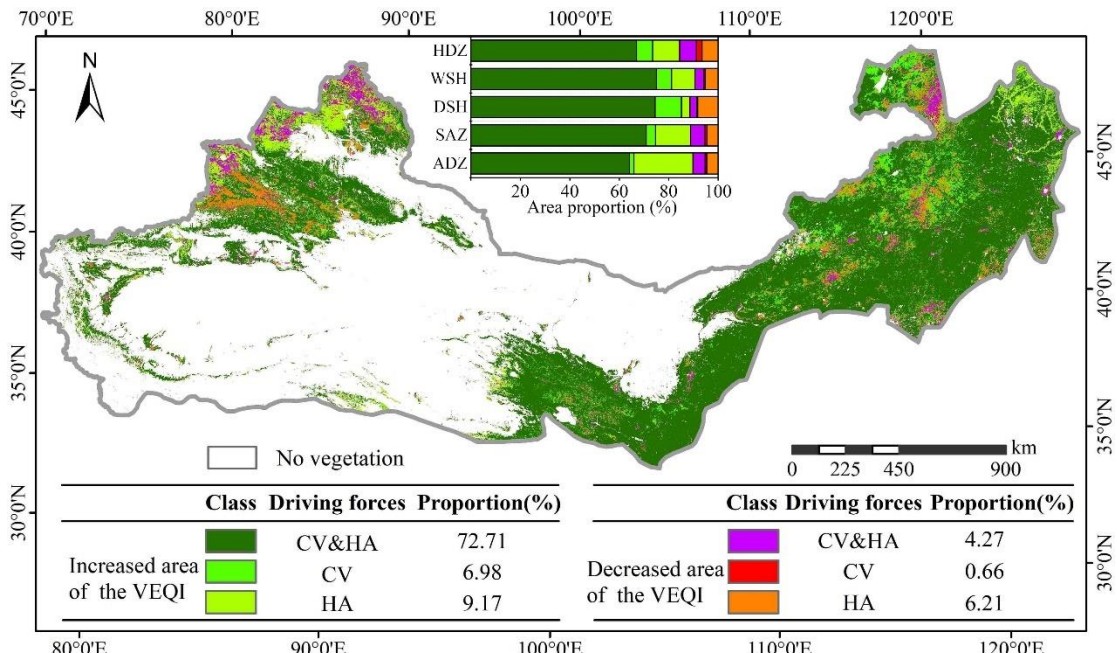

**Figure 12.** Spatial distribution characteristics of the drivers of VEQI changes in the TNS from 2000 to 2020.

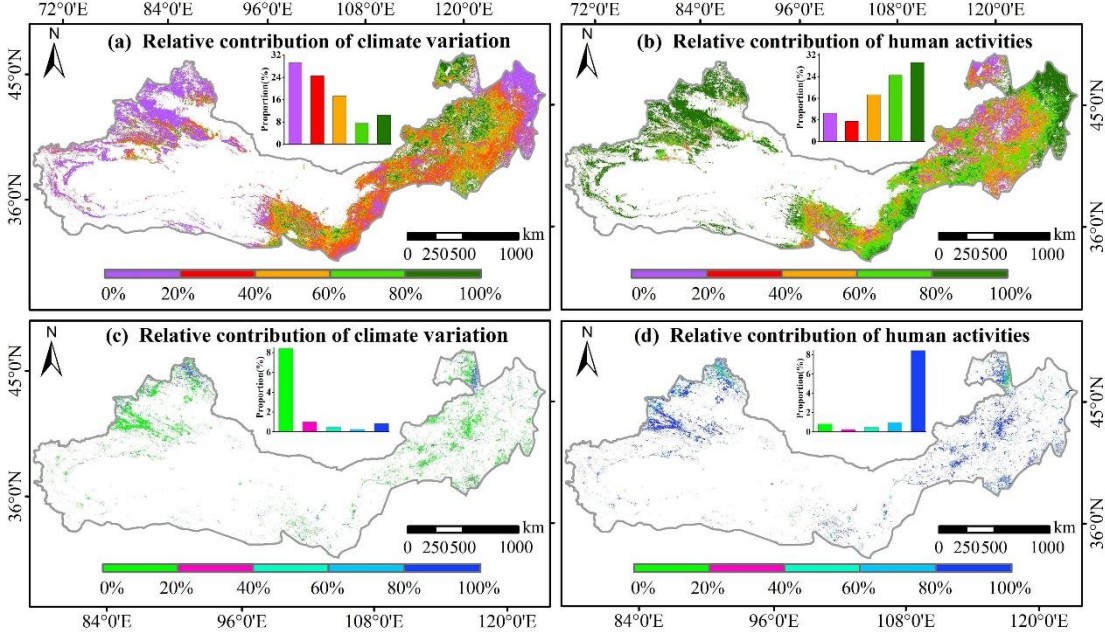

**Figure 13.** Relative contributions of climate variation and human activities to VEQI: (**a**,**c**) the relative contributions to VEQI of climate variation in increasing (decreasing) regions, respectively; (**b**,**d**) the relative contributions to VEQI from human activities in increasing (decreasing) regions, respectively.

In summary, the interaction between climate variation and human activities caused the change of VEQ in the TNS from 2000 to 2020, in which human activities dominated. The research has shown that the contribution of climate variation and human activities to VEQ varied considerably across climate zones (Figure S4). In the regions where VEQ improved, the largest contribution gap was in ADZ and DSH. Conversely, in the regions where VEQ deteriorated, the largest contribution gap was in DSH and HDZ.

## 4. Discussion

### 4.1. Spatial and Temporal Variation in VEQ

Many researchers indirectly demonstrated that the VEQ of the TNS has improved in the past 21 years. For example, Zhang et al. [8] found that several afforestation programs accelerated the greening of vegetation in the TNS from 1982 to 2013, based on NDVI data. Hu et al. [4] indicated that most of the LAI in the TNS showed an increasing trend during 2000–2015, based on GLASS LAI data. Zhang et al. [15] also discovered that GPP and WUE in the TNS displayed a small increasing trend during 2001–2017. The areas showing significant improvement in VEQ were primarily influenced by human activities (e.g., grain for green projects, returning farmland to grass, and planting trees). The areas with significant VEQ deterioration were mainly scattered in various urban areas, which was primarily the result of the conversion of vegetated areas into constructed areas due to the rapid development of the urbanization process.

The results of the Hurst exponent analysis indicate that about half of VEQ in the TNS was not sustainable in the future. This can be partly explained by the fact that the TNS is made up of mostly arid and semi-arid climates, and the vegetation quality has likely peaked, requiring serious consideration in the future. The survival and preservation rates of afforestation in arid windy and sandy areas are low, the quality of protected forests is poor, and the risk of decline is high [10]. In addition, the constraints of the carrying capacity of water resources were not fully considered at the beginning of the protective forest project. The implementation of the project has caused groundwater to decline, and tree growth will be severely affected in the future, likely leading to degradation and aging [10,55]. The future degradation of regions with improved VEQ can be prevented by adjusting the restoration strategy of these areas and including science-driven afforestation, fallowing, post-cultivation, and water connotation.

### 4.2. Drivers of VEQ

The factor with the highest contribution was PRE, which is consistent with the results of previous studies [56,57]. Climate is often considered to be a pivotal factor in vegetation growth. PRE is strongly associated with the diversity and amount of vegetation. Our results suggested that PRE had a significant contribution to the VEQ. The factor with the next highest contribution to VEQ was VT. Studies have shown that the VEQ varies greatly among VT. For example, coniferous and broad-leaved forests have better ecological quality than grassland and desert vegetation. In addition, our study showed that the remaining individual factors did not contribute much to the VEQ, but when interacting with other factors, they could increase the explanatory power. Results from previous researchers also indicated that the driver interactions were stronger than the individual factors [48,56].

The TNS has shown a warming and wetting trend from 2000 to 2020. Warming can accelerate the decay of soil organic substances and lengthen the vegetation growing period, thus improving VEQ. In the past decades, the government has implemented some measures (e.g., afforestation, reforestation, grassland restoration, and sandy land protection) which have significantly improved the VEQ and ecological benefits. However, climate variation and human activities can also cause the deterioration of the VEQ. The deterioration of the VEQ in the central Greater Khingan Mountains of China may be related to the decrease in TEM and PRE. The negative effects of human activities on the VEQ were concentrated in large urban agglomerations, mainly reflecting the encroachment on woodlands or grasslands by massive industrial activities and urbanization. In addition, the

study indicates that the effects of human activities on the VEQ were greater than those of climate variation. This finding was indirectly analogous to the studies of Huang et al. [50] and Sun et al. [58].

### 4.3. Afforestation, Forestry System Fixed Investment, and Land Use Transfer

We further explored the impact of human activities on VEQ in terms of afforestation, fixed investment in forestry systems, and land use transfer. The afforestation project controlled sandy arable land and water loss, which has greatly improved the VEQ in the TNS. We analyzed the annual and cumulative afforestation area in the TNS and each province based on statistical yearbook data (Figure 14). The correlation between the cumulative afforestation area and VEQI was also studied (Figure 15). The results suggested that the annual afforestation area in the TNS and its provinces was stable. The cumulative afforestation area of the TNS was $4.68 \times 10^5$ km$^2$. The coefficient between the cumulative afforestation area and the VEQI was 0.917, indicating that the improvement of the VEQ from 2000 to 2020 was closely associated with the afforestation projects organized and implemented by the government. In addition, we found that the correlation coefficients of the cumulative afforestation area and VEQI for Qinghai, Neimenggu, Beijing, Tianjin, and Liaoning were lower than 0.8. This may be due to the inherently high VEQI values in Beijing, Tianjin, and Liaoning, where the increase in cumulative afforestation area did not result in a significant increase in VEQI. In contrast, Qinghai and Neimenggu may have inherently lower VEQI values due to their vegetation cover types, which are dominated by desert and low-cover grassland. Although the cumulative afforestation area has increased, the VEQ has not improved significantly on a large scale due to the dry climate and low precipitation in these two provincial administrations, which are located mainly in arid and semi-arid areas.

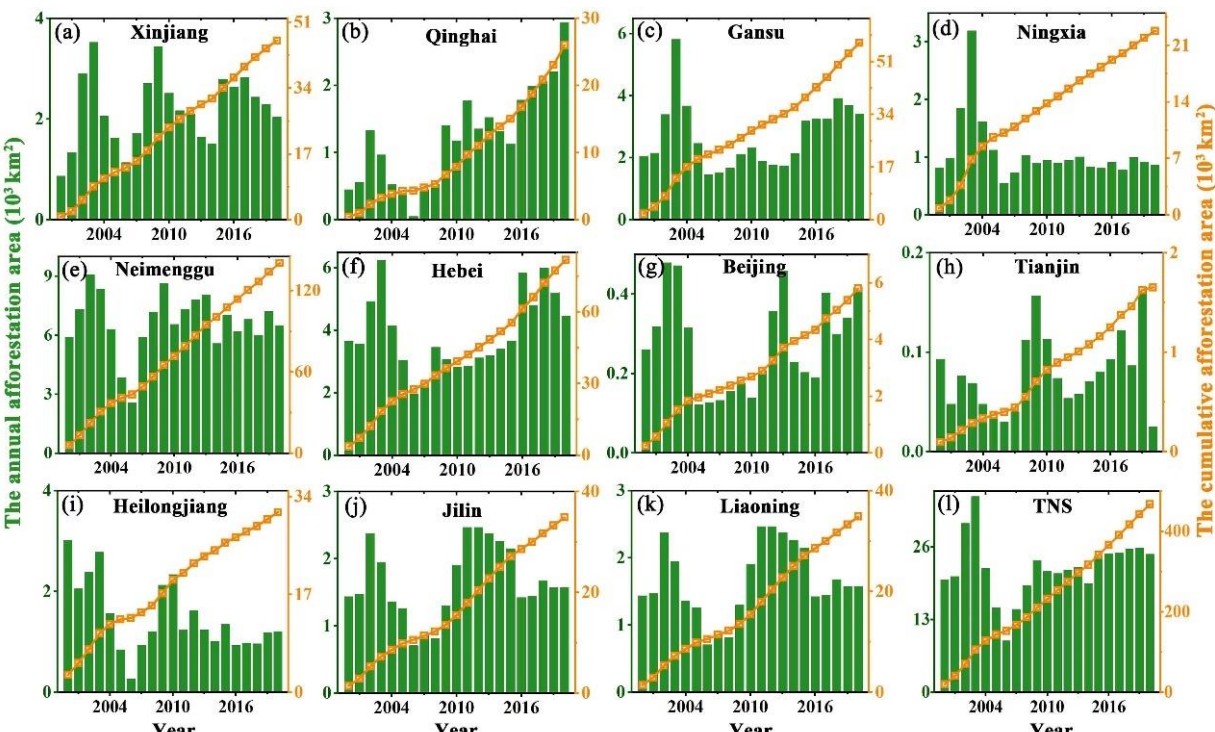

**Figure 14.** Statistics of plantation area and cumulative plantation area of the TNS (**l**) and each province (**a–k**) from 2000 to 2020.

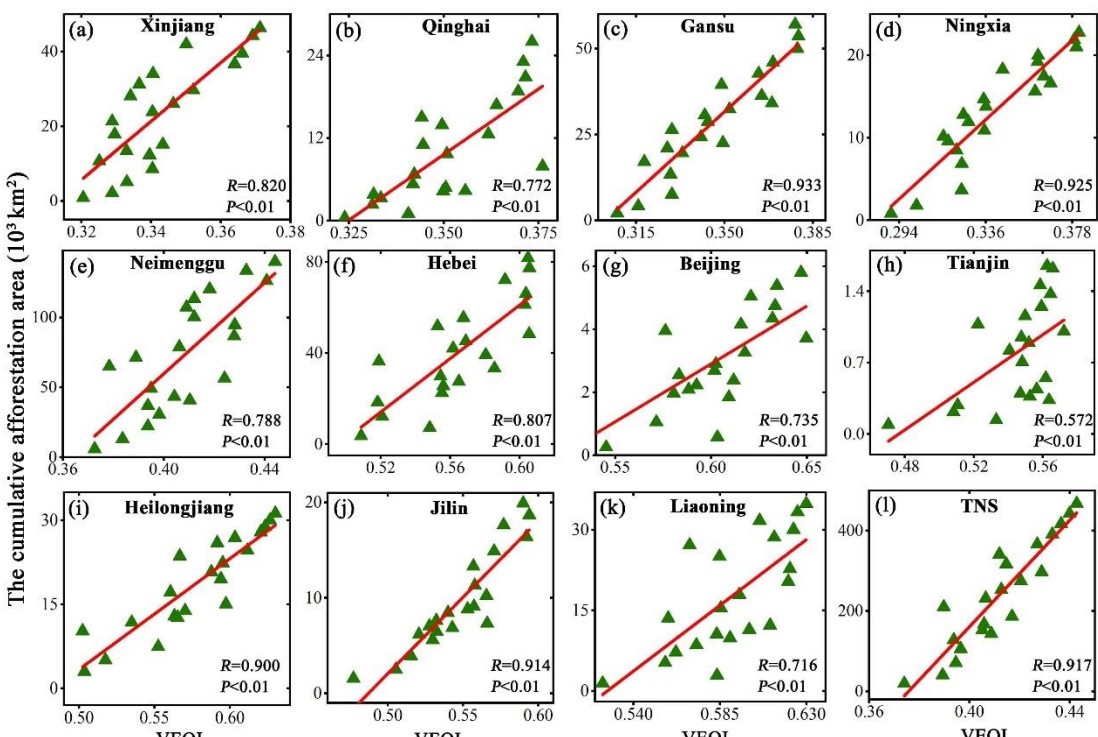

**Figure 15.** The relationships between the cumulative planted area and VEQI (**l**) in the TNS and each province (**a–k**).

We obtained the transfer characteristics of land use types based on the Markov transfer matrix. The main land use transfer types were grassland to cropland (69,656 km$^2$), barren to grassland (38,410 km$^2$), cropland to grassland (27,729 km$^2$), grassland to barren (7095 km$^2$), and grassland to forestland (5926 km$^2$) (Table S4, Figure S5). The land use types showing the greatest changes were cropland and barren land. The net increase in cropland was 40,226 km$^2$, mainly converted from grassland. The net decrease in barren land was 33,039 km$^2$, mainly due to a shift to grassland. In addition, the total vegetation cover region displayed an increasing trend, with a net increase of 30,678 km$^2$.

The implementation of environmental policies requires a certain amount of financial investment. Therefore, we analyzed the annual fixed and cumulative fixed investment in forestry ecological projects in the TNS and its provinces (Figure S6). The correlation between cumulative fixed investment and VEQI was also analyzed (Figure S7). The results reveal that the fixed investment in the forestry system in the TNS showed an increasing trend. Except for Liaoning Province, the annual fixed investment in forestry systems in the remaining provinces also showed an increasing trend, indicating the importance attached by the government to forestry restoration. The cumulative fixed investment in forestry systems in the TNS was $1.14 \times 10^{12}$ CNY from 2000 to 2020. The coefficient of cumulative fixed investment in the forestry system and VEQI for the TNS was 0.856.

Large-scale afforestation was the primary method used to increase the vegetation cover in the initial stage of environmental restoration of the TNS. With the implementation of the afforestation project, the land use types in the TNS have changed significantly. Specifically, low VEQ types shifted to higher VEQ types, such as grassland to cropland, forestland, and shrub. Non-vegetated areas were converted into vegetated areas (e.g., barren to grassland). This has slowed down, stopped, or even reversed the desertification process and VEQ deterioration in some areas [59]. These ecological restoration measures would not have been possible without fixed investments from the government. With strong government support, forestry investment steadily increased, enabling the engineering renovation project to continue effectively. In summary, national ecological restoration projects played a critical

role in improving the environmental quality of the TNS and promoting sustainable land development, and they were the main drivers of vegetation change.

### 4.4. Limitations and Prospects

We used a large amount of basic remote sensing data, such as NDVI, LAI, and GPP, in the construction of the VEQI. Although these vegetation indices are widely used due to their many advantages, such as easy accessibility, they also have many limitations and drawbacks. Firstly, the remote sensing data we used are optical images, and there will be missing values and anomalies affected by terrain shadows, thick clouds, and haze [6]. Although we performed operations such as invalid value filling, outlier processing, and Savitzky–Golay smoothing filtering on the downloaded remote sensing data, we also tried to avoid the effects of random values and some systematic errors by methods such as the maximum value composite procedure [60]. However, whether the processed data match the actual relevant feature values of the vegetation must be verified with the help of field observation data. Secondly, we used NDVI data to calculate FVC. When the ground vegetation becomes denser, the NDVI becomes saturated, and the simultaneous growth can not be achieved [61]. NDVI is also susceptible to disturbance by the soil background, especially in areas with medium vegetation cover. NDVI tends to increase when the soil background becomes darker [61]. Finally, the use of NDVI may relate mainly to the vegetation growing season of the area. Since NDVI is related to the amount of vegetation structure and photosynthetic biomass [62], it is not valid for images taken during the vegetation growing season, when vegetation is reduced, or during the non-vegetation growing season.

We used VEQI constructed from remote sensing data for evaluating the VEQ in the TNS. VEQI was mainly generated by referring to the relevant index factors, core technical principles, and the framework of RSEI proposed by Xu. [18], and by integrating six basic indexes that were widely used and more important in vegetation ecology assessment for dimensionality reduction. The core techniques involved in VEQI (i.e., data normalization processing and PCA) are similar to those of RSEI, which indicates that the technical principles used in VEQI are reliable. Moreover, the results of the PCA suggested that the VEQI we used and constructed was reasonable. Therefore, the VEQI we used can be used to evaluate the VEQ in the TNS. However, there are still some limitations in our study of VEQI, such as a lack of comparative analysis of the VEQI values inferred from remote sensing data and the in situ observations. Therefore, in future work, we will conduct field measurements to obtain accurate measurement data for VEQI validation and improve the accuracy of VEQI.

We quantified the impact and relative contribution of climate change and human activities on VEQI using residual analysis. However, relevant studies have shown that vegetation activity can also be affected by extreme climatic phenomena such as extreme drought, El Niño, and La Niña [63]. This cannot be quantified by the residual analysis method. Therefore, we will assess the impact of extreme climate events on VEQI in a follow-up study.

### 4.5. Implications

Vegetation is a major source of carbon sequestration in terrestrial ecosystems [64]. For the past 20 years, terrestrial ecosystems in China have been important carbon sinks (0.19–0.26 Pg C/yr compared to 1–4 Pg C/yr for global terrestrial ecosystems, [65]), reducing the country's fossil fuels by more than 20% of [66]. Improving the carbon sink capacity of terrestrial ecosystems by building national ecological projects (afforestation and reforestation, etc.) is a significant pattern for achieving carbon neutrality, which can greatly mitigate the effects of regional carbon emissions [67].

Our study suggested that the VEQ has greatly increased from 2000 to 2020. Considering its large area, it indicates that the TNS probably became a large carbon sink. The ecosystem services of TNS must include the carbon sink when evaluating the impact of the TNS project, which has been limitedly reported previously.

Our study demonstrates that Chinese afforestation projects are the main drivers of regional environmental conservation, indicating that artificial interventions in ecologically fragile areas are effective. These benefits could be gained in other ecologically fragile areas around the world if they were improved by a long-term ecological recovery project, which can benefit adaptation to and mitigation of global warming under the Paris agreement.

On the other hand, the VEQ in some regions has probably reached the tipping point and may decrease in the future. This indicates that the local natural environment and plant water consumption patterns should be considered for different climatic zones to avoid unreasonable vegetation management strategies. In particular, reasonable ecological restoration and nurturing measures should be adopted in arid and semi-arid zones, with gradual and comprehensive management. Ecosystem recovery projects in arid and semi-arid regions must consider local environmental conditions. Related research should be carried out first, which can help balance the budget and benefit other fragile regions.

## 5. Conclusions

We studied the spatial and temporal trends, future change characteristics, and main driving mechanisms of the VEQ in the TNS by constructing the VEQI. The results indicated that the VEQ in the TNS has been greatly improved. Significantly improved areas were concentrated in the central and eastern parts of the TNS. Significantly deteriorated areas were not clustered in large areas and were primarily scattered in various urban areas. The Hurst exponent indicated that the future trend of VEQ in the TNS is unsustainable. PRE, VT, ST, elevation, and SNR had the greatest effect on the VEQ across the TNS. PRE and SNR were linearly and positively correlated with VEQI, while elevation and ASP were linearly and negatively correlated with VEQI. The interaction of climate variation and human activities influenced the variation of the VEQ in the TNS. Human activities (afforestation and fixed government investments, etc.) were the dominant factor, with a relative contribution of 65.24% (34.76% for climate variation).

**Supplementary Materials:** The following supporting information can be downloaded at: https://www.mdpi.com/article/10.3390/rs14225708/s1, Text S1: Principal component analysis of each indicator. Figure S1: Area proportion statistics of VEQI for each grade over the years. Figure S2: Spatio-temporal change characteristics of TEM (a,d), PRE (b,e), and SNR (c,f) in the TNS from 2000 to 2020. Figure S3: Temporal trends in the residuals of TNS and each climate zone. Figure S4: Statistics of the relative contributions of climate variation and human activities to VEQI in each climate zone: (a) and (c) are the relative contributions of VEQI to climate variation in increasing (decreasing) regions, respectively; (b) and (d) are the relative contributions of human activities to VEQI in increasing (decreasing) regions, respectively. Figure S5: Spatial distribution of the transfer of main land-use types of the TNS from 2000 to 2020. Figure S6: Statistics of annual fixed and cumulative fixed investments in forestry projects in the TNS and each province from 2000 to 2020. Figure S7: The relationships between cumulative fixed investment and VEQI of forestry projects in the TNS and each province. Table S1: Results of PCA of each indicator. Table S2: Future trends of VEQ in the TNS. Table S3: Suitable scopes or types of the factors. Table S4: Statistics for the proportion of land use transfer area in the TNS from 2000 to 2020 (km$^2$).

**Author Contributions:** Conceptualization, C.L. and S.Z.; methodology, C.L.; software, M.C.; validation, M.C., X.W. and J.W.; formal analysis, T.R.; resources, W.L.; data curation, C.L.; writing—original draft preparation, C.L.; writing—review and editing, S.Z.; visualization, C.L.; supervision, S.Z.; project administration, S.Z.; funding acquisition, S.Z. All authors have read and agreed to the published version of the manuscript.

**Funding:** This research was supported by the National Natural Science Foundation of China (Grant No. 41730751).

**Data Availability Statement:** Not applicable.

**Acknowledgments:** We would like to acknowledge the anonymous reviewers and editors whose thoughtful comments helped to improve this manuscript.

**Conflicts of Interest:** The authors declare no conflict of interest.

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
