# Peer review of "Improved Vegetation Ecological Quality of the Three-North Shelterbelt Project Region of China during 2000–2020 as Evidenced from Multiple Remotely Sensed Indicators"

_remotesensing, doi:10.3390/rs14225708_

Round 1

Reviewer 1 Report

The manuscript is well written, there are some little problems need to be solved.

Line 56: evapotranspiration (EVA) -- potential evapotranspiration (EVA)?

Line 114: Please give more description about “outlier processing”, especially for Land Surface Temperature.

Table 1: These products are used for which evaluation indexes, e.g. FVC, WUE, …

Line 121: Please add the DOI of the dataset.

Line 153: Please give more description about WET and WUE.

Line 289: TEM TEM, TEM maybe DEM ?

Section 2.3.3. Residual analysis: How to consider the impact of extreme climate events, such as drought, can be discussed a little more.

Reviewer 2 Report

In this study, China's Three North Shelter Forest was used as the study area, and a comprehensive vegetation ecological quality index (VEQI) was constructed to evaluate the vegetation status through principal component analysis. The variation trend of VEQI was analyzed by Sen's slope analysis, M-K test and Hurst index, and driving factor analysis was also performed by geographic detector and residual analysis. The results are basically satisfactory, but there are still some questions that need to be revised and answered by the author.

1. Regarding the remote sensing data in Table 1, did the authors use the annual average of each remote sensing data? The author should make it clear in the article.

2. Lines 115 and 116. The authors indicated that water bodies, snow cover and bare soil were masked. How is the mask done? Is it through land cover data?

3. Why should two methods be used to explore the influences of VEQI? It is well known that geodetector models are commonly used to explore the drivers, why does this paper also use residual analysis to explore the effects of climate change and human activities on VEQI separately?

4. The author uses principal component analysis to combine FVC, LAI, NPP, WET, LST and WUE 6 factors to construct VEQI, but the process of principal component analysis is not mentioned in the article. It is recommended to list the contribution table of each principal component of the principal component analysis

5. The construction of RSEI by principal component analysis by Xu has been widely used and various improved RSEIs have emerged. Some time ago Xu also posted an article to criticize that some improved RSEIs are not reasonable. How can the authors ensure that their VEQI constructed by principal component analysis is reasonable and whether VQEI is really more responsive to vegetation condition than FVC? None of these authors have argued.

6. When the authors performed the geographic probe and residual analysis for driving factor analysis, did they use the average of a particular year for each data? Or is it the average of all data? It is not clear from the authors.

7. In the study area, what is the basis for classifying the arid zone (ADZ), semi-arid zone (SAZ), dry semi-humid zone (DSH), wet semi-humid zone (WSH), and humid zone (HDZ)? Please include a reference or add a description.

8. From the Figure 15 we can clearly see that the relationships between the cumulative planted area and VEQI in Beijing,Tianjing,Liao ning and Neimenggu and the TNS and each province is not high(R lower than 0.8). There is no explanation why relationships between the cumulative planted area and VEQI in those area?   

9. There are some minor problems. The keywords do not reflect the central word of this paper, "vegetation ecology quality”. In Figure 11, changes are made to the shaded portions of the text. In Figure 13, the map scale is missing.

Reviewer 3 Report

The relevance of the research topic is related to projects to restore the ecosystems of the region for the sustainable development of territories and environmental safety. An important role in the scientific solution of this problem is assigned to the interpolation of statistical data of meteorological measurements and observations of plant ecosystems and their spatial distribution obtained using satellite images for territories with a characteristic pronounced uneven relief of urban development and microclimate. From these positions, the article considers the spatial and temporal characteristics and the main factors for assessing the state of plant ecosystems. For these purposes, Earth remote sensing technologies using Landsat 8 OLI satellite image channels are used (http://landsat.usgs.gov ). Landsat data was an integral component of this work, both for mapping vegetation cover. The use of the NDVI index becomes especially significant when calculating productivity, biomass reserves and other quantitative indicators. As practice shows, this is a very commonly used approach for preparing data for creating NDVI maps. The main advantage of vegetation indices is the ease of obtaining them and a wide range of tasks solved with their help. However, there are a number of disadvantages and limitations.

Disadvantages of using NDVI-index, which can be emphasized in the article:

 the need for most tasks to compare the results obtained with the pre-collected data of test sites (standards), which should take into account seasonal environmental and climatic indicators, both of the image itself and of test sites at the time of data collection;

 errors introduced by weather conditions, heavy clouds and haze - their influence can be partially corrected using improved coefficients and composite images with NDVI series for several days, weeks or months (MVC - Maximum Value Composite). Averaged values allow avoiding the influence of random and some systematic errors.

Another significant disadvantage of using the NDVI index is the possibility of using only the time of the growing season for the region under study. Due to its attachment to the amount of photosynthetic biomass, NDVI is not effective on images taken during the season of weakened or non-vegetating vegetation during this period.

As a recommendation to the authors, refer to the following article

Zhao, Feng; Meng, Ran; Huang, Chengquan; Zhao, Maosheng; Zhao, Feng; Gong, Peng; Yu, Le; Zhu, Zhilian (October 29, 2016). "Long-term forest restoration after disturbance in the Greater Yellowstone ecosystem, analyzed using the Landsat time series stack." Remote sensing. MDPI AG. 8 (11): 898. Bibcode: 2016RemS....8..898Z. doi:10.3390/rs8110898. ISSN 2072-4292.

Round 2

Reviewer 2 Report

The revision made significant improvements according to the comments. I believe that the authors have substantially improved the content and form of the article and it can now be published in the journal.